# Biodiversity and Safety: Cohabitation Experimentation in Undefined Starter Cultures for Traditional Dairy Products

Luigi Chessa *, Elisabetta Daga, Ilaria Dupré, Antonio Paba, Maria C. Fozzi, Davide G. Dedola and Roberta Comunian

Agris Sardegna, Servizio Ricerca Prodotti di Origine Animale, Associated Member of the JRUMIRRI-IT, Loc. Bonassai SS 291 km 18.600, 07100 Sassari, Italy; edaga@agrisricerca.it (E.D.); idupre@agrisricerca.it (I.D.); apaba@agrisricerca.it (A.P.); mfozzi@agrisricerca.it (M.C.F.); dgdedola@agrisricerca.it (D.G.D.); rcomunian@agrisricerca.it (R.C.)
* Correspondence: lchessa@agrisricerca.it

**Abstract:** Natural starter cultures, characterised by undefined microbiota, can contribute to the technological process, giving peculiar characteristics to artisanal fermented foods. Several species have a long history of safe use and have obtained Qualified Presumption of Safety (QPS) status from the European Food Safety Authority (EFSA), whereas others (non-QPS) could represent a potential risk for consumers' health and must undergo a safety assessment. In this work, the biodiversity, at species and strain level, by pulsed-field gel electrophoresis (PFGE) and (GTG)$_5$ rep-PCR, of an undefined natural starter culture, in frozen and lyophilized form, obtained from ewe's raw milk avoiding thermal treatment or microbial selection, was investigated. The culture was constituted by different biotypes of *Enterococcus durans*, *Enterococcus faecium*, *Enterococcus faecalis*, and *Lacticaseibacillus paracasei*. *Streptococcus oralis* and *Streptococcus salivarius* were also found, over species belonging to the *Streptococcus bovis*–*Streptococcus equinus* complex (SBSEC), like *Streptococcus gallolyticus* subsp. *macedonicus*, *Streptococcus lutetiensis*, and *Streptococcus equinus*. Molecular investigation on virulence and antibiotic resistance genes, as well as minimum inhibitory concentration (MIC) determination, revealed that all the non-QPS strains can be considered safe in the perspective of using this culture for cheesemaking. The obtainment of a natural culture directly from ewe's raw milk bypassing thermal treatment and selection of pro-technological bacteria can be advantageous in terms of biodiversity preservation, but non-QPS microorganisms can be included in the natural starter and also in cheeses, especially in traditional ones obtained from fermenting raw milk. Following EFSA guidelines, artisanal factories should not be allowed to produce starter cultures by themselves from raw milk, running the risk of including some non-QPS species in their culture, and only selected starters could be used for cheesemaking. A revision of the criteria of QPS guidelines should be necessary.

**Keywords:** natural starters; cheesemaking; food safety; biodiversity; microbial fingerprint

## 1. Introduction

Starter cultures are used to aid raw material processing with the aim of easily and safely carrying out fermentation and obtaining different types of fermented foods. In cheesemaking, commercial starters consisting of a few (one to three) selected species/strains, or natural starter cultures, can be used [1,2]. Selected starters represent a suitable solution to perform fermentation if the microbiota of raw material and production environment is inadequate, or natural starter cultures are difficult to obtain and manage [3]. They are widely added at high concentrations in industrial production processes, becoming dominant in the food microbiota, causing a dramatic decrease in microbial diversity and a loss of peculiar sensory characteristics of foods of particular geographic niches [4]. On the contrary, natural cultures are complex microbial communities that, having a strain composition mostly undefined [5], are not reproducible in any place other than their origin.

Their use in food production contributes to preserving microbial biodiversity, enriching artisanal products with peculiar sensory features that link them to the territory of production [6]. Indeed, autochthonous natural starter cultures usually characterise the most typical and high-quality agri-food products. Natural cultures are usually reproduced daily by cheesemakers, but after repetitive passages of reproduction, they are not able to properly accomplish their technological role (i.e., acidification ability) any longer.

The use of these natural cultures could not be risk free since, together with useful autochthonous microorganisms, even pathogenic or spoilage ones could be potentially inoculated and can contaminate the product [2]. Several microorganisms can be used for food production, such as lactic acid bacteria, whereas others could represent a potential risk for the consumers' health, e.g., enterococci, some streptococci, and staphylococci. In recent decades, the European Food Safety Authority (EFSA) introduced the definition of Qualified Presumption of Safety (QPS) for long-safe-history microorganisms that can be used for food and feed production without prior safety assessments [7]. The QPS list is published every three years, and the updates are based on a two-year assessment carried out by the Biological Hazards (BIOHAZ) Panel [8]. The QPS status is the result of a pre-assessment that covers safety concerns for humans, animals, and the environment. During this process, experts assess the taxonomic identity of the microorganism and the related body of knowledge. Microorganisms that are not well defined, and for which it is not possible to conclude whether they pose a safety concern, are not considered suitable for QPS status and must undergo a full safety assessment. However, also some non-QPS microorganisms can be used after the ascertainment of safety, e.g., *Enterococcus faecium* [9,10]. A possible solution to overcome these problems (obtainment, management, safety, and technological ability loss) could be the use of natural cultures in lyophilized form [2].

The aim of this work was to evaluate the biodiversity, at species and strain level, of an undefined natural starter culture, obtained from raw ewe's milk, avoiding any thermal treatment or microbial selection, as well as to assess the safety of non-QPS strains in the perspective of use of this culture, in frozen or lyophilised form, for cheesemaking.

## 2. Materials and Methods

### 2.1. Experimental Plan

A natural starter culture for cheesemaking, in frozen (NSC) and lyophilized (LNSC) form, was characterised for its microbial diversity, at the strain level, and safety. NSC was obtained directly from raw ewe milk in a study described by Chessa et al. [11], where the composition in microbial groups (i.e., mesophilic cocci and bacilli, thermophilic cocci and bacilli, and enterococci) and the technological performances in acidification were assessed. Moreover, NSC was lyophilized by Veneto Agricoltura (Thiene, Italy) as external service. In this study, both the frozen (NSC) and lyophilized (LNSC) forms of the natural starter culture were characterised for their microbial diversity. Moreover, the evaluation of safety for non-QPS bacteria isolated from NSC and LNSC was also performed. Operatively, microbial counts in NSC, after thawing, and LNSC, after lyophilisation, were performed, and a total of 157 microbial colonies were picked up from the different elective media used for the enumeration of microbial groups. The microbial isolates were identified, first, by species-specific PCR and then molecularly typed by (GTG)$_5$ rep-PCR or pulsed-field gel electrophoresis (PFGE). Molecular and phenotypic tests for the detection of antibiotic resistance and virulence genes, and for the determination of minimum inhibitory concentration (MIC), were also performed.

### 2.2. Microbial Counts and Isolation

Microbial counts on both NSC and LNSC for the enumeration of thermophilic cocci and lactobacilli, mesophilic cocci and lactobacilli, enterococci, citrate-fermenting bacteria, staphylococci, and coliforms were performed by spreading 0.1 mL of each serial 10-fold dilution on agar plates, as described by Chessa et al. [11]. From each media used for microbial counts, 10 colonies were picked up from the lowest countable dilution (Table S1), and

each colony was purified by 3 repetitive passages on the agar media of origin. Morphology of isolates was also checked using Axio-Phot optic microscope (Carl Zeiss, Oberkochen, Germany) equipped with Objective EC Plan-Neofluar 1009/1.30 OilPol M27.

### 2.3. Microbial Identification and Biodiversity Evaluation

#### 2.3.1. Species Identification

The identification of 157 bacterial isolates was performed by genus/species-specific PCR and DNA sequencing analysis. Genomic DNA was extracted from purified cocci and bacilli, isolated from NSC and LNSC, using PrepMan Ultra Sample Preparation reagent (Applied Biosystems—Thermo Fisher Scientific, Rodano, Italy), according to the manufacturer's instructions. The extracted DNA was diluted 1:50 in sterile Milli-Q water and used as template for genus/species determination with the primers listed in Table 1. PCR products were separated on agarose gel, supplemented with 1× SYBR SAFE (Invitrogen—Thermo Fisher Scientific) in Tris-acetate buffer, and gel images were acquired by the UV transilluminator FireReader V4 (UVITec, Warwickshire, UK). The microbial isolates for which it was not possible to identify by genus/species-specific PCR were investigated via DNA sequencing analysis targeting 16S rRNA and *sod*A genes using the universal primers *p27f* and *p765r* [12], and *d1* and *d2* [13] targeting *16S rRNA* (about 750 bp) and *sodA*, respectively (Table 1). Sequences were edited and aligned with BioEdit (v. 7.2) using the ClustalW algorithm. Consensus sequences were built for each sample and, for species determination, were compared to those deposited in the nucleotide database of the National Center for Biotechnology Information by means of NCBI BLAST. Identification was deemed reliable if values for sequence similarities were >99%.

**Table 1.** Molecular primers used for microbial identification.

| Genus/Species | Primer | Gene Target | Primer Sequence (5′–3′) | Annealing (°C) | Size (bp) | Reference |
|---|---|---|---|---|---|---|
| *Lactococcus lactis* | LcLspp-F | *16S rRNA* | GTTGTATTAGCTAGTTGGTGAGGTAAA | 55 | 387 | [14] |
| | Lc-R | | GTTGAGCCACTGCCTTTTAC | | | |
| *Lactobacillus delbrueckii lactis* | Lac-LACTIS-F733 | *dppE* | TGCCAAGCTCTACTCCGTTT | 58 | 217 | [15] |
| | Lac-LACTIS-R949 | | GTCAAGCGGCATAGTGTCAA | | | |
| *Lactobacillus delbrueckii bulgaricus* | Lac-BULG-F391 | *lacZ* | GGAAGACTCCGTTTTGGTCA | 58 | 395 | [15] |
| | Lac-BULG-R785 | | AGTTCAAGTCTGCCCCATTG | | | |
| *Lactobacillus helveticus* | Lac-HELV-F73 | *prtH* | GGCGGGGAAAGAGGTAACTA | 58 | 509 | [15] |
| | Lac-HELV-R581 | | TGACGCAAACTTAATGAACCA | | | |
| *Limosilactisbacillus fermentum* | Lac-FER-F753 | *ArcD* | CCAGATCAGCCAACTTCACA | 58 | 310 | [15] |
| | Lac-FER-R1062 | | GGCAAACTTCAAGAGGACCA | | | |
| *Limosilactobacillus reuteri* | REUT1 | *16S rRNA* | TGAATTGACGATGGATCACCAGTG | 65 | 1000 | [16] |
| | LOWLAC | | CGACGACCATGAACCACCTGT | | | |
| *Lacticaseibacillus paracasei* | Y2 | *16S rRNA* | CCCACTGCTGCCTCCCGTAGGAGT | 55 | 290 | [17] |
| | PARA | | CACCGAGATTCAACATGG | | | |
| *Lactiplantibacillus plantarum* | planF | *recA* | CCGTTTATGCGGAACACCTA | 56 | 318 | [18] |
| | pREV | | TCGGGATTACCAAACATCAC | | | |
| *Streptococcus thermophilus* | Str-THER-F2116 | *lacZ* | GCTTGTGTTCTGAGGGAAGC | 58 | 577 | [15] |
| | Str-THER-R2693 | | CTTTCTTCTGCACCGTATCCA | | | |
| *Enterococcus* | ENT1 | *Tuf* | TACTGACAAACCATTCATGATG | 59 | 112 | [19] |
| | ENT2 | | AACTTCGTCACCAACGCGAAC | | | |
| *Enterococcus faecium* | FM1 | *sodA* | GAAAAAACAATAGAAGAATTAT | 55 | 215 | [20] |
| | FM2 | | TGCTTTTTTGAATTCTTCTTTA | | | |

**Table 1.** *Cont.*

| Genus/Species | Primer | Gene Target | Primer Sequence (5′–3′) | Annealing (°C) | Size (bp) | Reference |
|---|---|---|---|---|---|---|
| *Enterococcus faecalis* | FL1 | *sod*A | ACTTATGTGACTAACTTAACC | 55 | 360 | [20] |
| | FL2 | | TAATGGTGAATCTTGGTTTGG | | | |
| Universal | p27f | *16S rRNA* | GAGAGTTTGATCCTGGCTCAG | 58 | ≈750 | [12] |
| | p765r | | CTGTTTGCTCCCCACGCTTTC | | | |
| Degenerate | d1 | *sod*A | CCITAYICITAYGAYGCIYTIGARCC | 37 | ≈480 | [13] |
| | d2 | | ARRTARTAIGCARRTARTAIGCRTGYTCCCAIACRTC | | | |

N = A, C, G, and T; R = A and G; W = A and T; Y = C and T.

2.3.2. Molecular Biotyping

The intraspecific identification of all the bacteria isolated from NSC and LNSC (*Enterococcus durans*, *Enterococcus faecium*, *Enterococcus faecalis*, *Streptococcus gallolyticus* subsp. *macedonicus*, *Streptococcus equinus*, *Streptococcus lutetiensis*, *Streptococcus oralis*, *Streptococcus salivarius*, and *Lacticaseibacillus paracasei*) was performed by (GTG)$_5$ rep-PCR microbial fingerprint, as described by Chessa et al. [21], using an FTA$^®$ Disc for DNA analysis (GE Healthcare, Chicago, IL, USA) as template. PCR products were separated on agarose gel (1.8% *w/v*) with 1× SYBR Safe (Invitrogen—Thermo Fisher Scientific), at 100 V (222 V/h) in Tris-acetate buffer.

Only the isolates identified as *S. equinus* and *S. lutetiensis* were genotyped by pulsed-field gel electrophoresis (PFGE). Genomic DNA extraction was prepared according to the protocol described by Graves and Swaminathan [22], then digested with 25 U of *Sma*I (BioLabs, Heidelberg, Germany) for 4 h at 25 °C. Electrophoresis was carried out in a contour-clamped homogeneous electric field (CHEF)-Mapper apparatus (Bio-Rad Laboratories, Hercules, CA, USA) at 14 °C in 0.5× TBE buffer. DNA fragments were separated at 6 V/cm gradient with 120° angle for 16 h. The running time was divided into 3 blocks: block 1 of 5 h, initial switch time 1 s final switch time 20 s; block 2 of 5 h, initial switch time 1 s final switch time 5 s; block 3 for 6 h, initial switch time 10 s final switch time 40 s. Gels were stained with 1× SYBR Safe.

Gel images of both (GTG)$_5$ rep-PCR and PFGE gels were acquired with the UV transilluminator FireReader V4 (UVITec) and elaborated by BioNumerics (v. 6.6.11; Applied Maths, Sint-Martens-Latem, Belgium). Cluster analysis was performed by unweighted pair group method with arithmetic averages (UPMGA); then, Pearson and Dice similarity correlation indexes for (GTG)$_5$ rep-PCR and PFGE profiles, respectively, were used. The isolates sharing ≥93% similarity among the (GTG)$_5$ rep-PCR profiles and 100% similarity among the PFGE profiles analysed according to Tenover et al. [23] were considered the same biotype.

*2.4. Safety Assessment*

2.4.1. Antibiotic Resistance and Virulence Gene Detection

For safety assessment, PCR reactions were carried out using primers and annealing temperatures listed in Table S2. Total-community DNA from NSC and LNSC, extracted following the protocol described by Paba et al. [24], and DNA from isolates, previously extracted for microbial identification, were used as template. PCR products were separated on 1.5% (*w/v*) agarose gel with 1× SYBR Safe in Tris-acetate buffer.

For the assessment of safety of *E. faecium* isolated from the cultures, the detection of the pathogenic-related genes $hyl_{Efm}$, *esp*, and IS*16* was carried out, according to the European Food and Safety Authority (EFSA) guidelines [10]. Same protocol was applied for the *E. durans* and *E. faecalis* isolates. *E. faecalis* was also tested for the presence of tetracycline resistance genes *tet*M, *tet*K, *tet*L, *tet*S.

The *Streptococcus* isolates were checked for antibiotic resistance genes for macrolides (*ermA*, *ermB*, *ermC*, and *mefA*), lincosamides (*lnuC*), tetracyclines (*tet*M, *tet*K, *tet*L, *tet*S, and

the putative conjugative Tn*916*-like transposon), and for the presence of potential virulence genes: *scpB*, *hylB*, *bca*, *bac*, *emm*, *smeZ*, *speA*, *speG*, and *ssa*.

The natural starter cultures NSC and LNSC *in toto* were tested for all the genes listed above.

### 2.4.2. Antibiotic Susceptibility

The isolates belonging to *Enterococcus* and *Streptococcus* genera, not currently included in the QPS list of EFSA [9], were phenotypically tested, by broth micro-dilution method, for their susceptibility to several antibiotics. Ampicillin (Sigma Aldrich, Milan, Italy) minimum inhibitory concentration (MIC) determination for *E. faecium* was performed using homemade trays prepared according to the ISO 20776-1:2019 [25], and the MIC breakpoint $\leq 2$ mg/L indicated by EFSA for safety evaluation was applied [10]. For each tray, 7 *E. faecium* strains were tested. For each strain, positive (strain DSM 2570, equivalent to ATCC 29212) and negative (not inoculated) control wells were also included. Plates were inoculated with 100 µL of Mueller-Hinton Broth (Thermo Fisher Scientific, Rodano, Italy) at $1 \times 10^5$ CFU/mL final concentration, then incubated at 37 °C overnight before visual examination of microbial growth.

The *E. durans* (from LNSC) and *E. faecalis* (1 from NSC and 2 from LNSC) isolates were tested for antimicrobial susceptibility by the Sensititre™ EU Surveillance Enterococcus EUVENC AST Plate (Thermo Fisher Scientific).

For *Streptococcus* isolates, Sensititre™ STP6F (Thermo Fisher Scientific) plates for MIC determination were used. Operatively, 100 µL of reconstitution volume in Cation-Adjusted Mueller-Hinton Broth with Lysed Horse Blood (Thermo Fisher Scientific) at $1 \times 10^5$ CFU/mL final concentration of the well was used, and although the plate was intended for 50 µL, the resulting dilutions were twice the lower dilution. The antibiotics used in this study for MIC determination are listed in Table 3.

The interpretation of MIC breakpoints was based, for *Enterococcus*, on the Clinical and Laboratory Standards Institute (CLSI) [26], and for *Streptococcus*, on The European Committee on Antimicrobial Susceptibility Testing [27] (EUCAST), applied for most of the antibiotics tested, except for tetracycline, erythromycin, and chloramphenicol, for which the breakpoints indicated in Clinical and Laboratory Standards Institute (CLSI) [26] were considered (Table S3).

### 2.5. Statistical Analysis

Differences in microbial counts about the concentrations of the microbial groups investigated between NSC and LNSC were compared using the Student *t* test using the software SPSS Statistics (v. 21.0; IBM Corp., Armonk, NY, USA). Molecular fingerprints performed by (GTG)$_5$ rep-PCR and PFGE were elaborated by BioNumerics (v. 5.0; Applied Maths, Sint-Martens-Latem, Belgium). Cluster analysis for PFGE and (GTG)$_5$-rep fingerprints was performed using deep significance clustering (DICE) and Pearson's correlation index through the unweighted pair group method using arithmetic averages (UPGMA), respectively.

The Simpson's diversity index (DI) was calculated for each microbial species found in NSC and LNSC, where the number of biotypes was $\geq 2$, according to Hunter and Gaston [28], using Equation (1), where $N$ is the total number of isolates, $S$ the total number of biotypes identified, and *nj* the number of isolates belonging to each biotype.

$$DI = 1 - \frac{1}{N(N-1)} \sum_{j=1}^{S} nj(nj-1) \tag{1}$$

## 3. Results

### 3.1. Microbial Counts of the Natural Starter Cultures

Microbial counts performed on the starter culture object of this study (NSC) and its lyophilized form (LNSC) revealed that the concentration of viable cells in LNSC was always

1.2–2.2 Log CFU/g higher than in NSC (Figure 1). Indeed, mesophilic and thermophilic cocci, mesophilic and thermophilic bacilli, as well as enterococci, were always significantly ($p < 0.05$) higher in LNSC. Staphylococci and coliforms were not detected in NSC nor in LNSC.

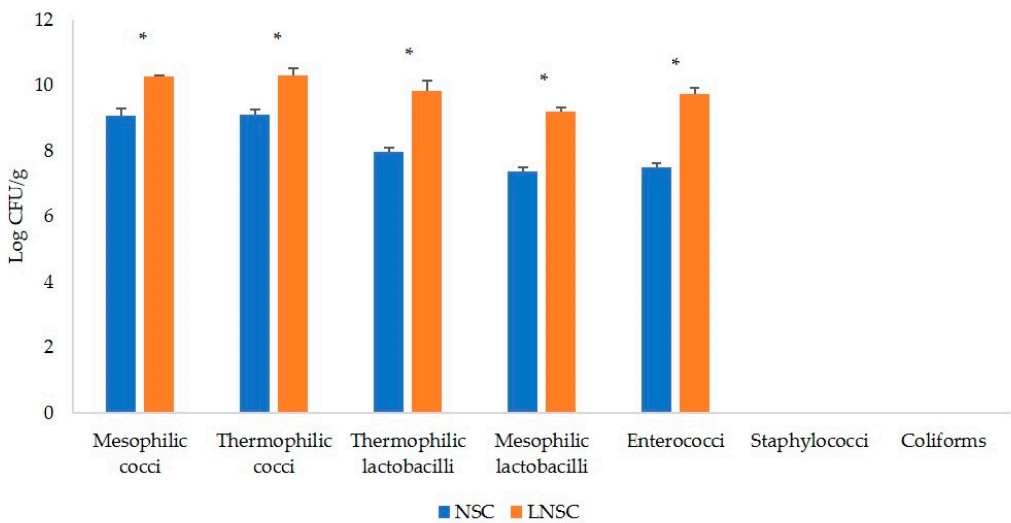

**Figure 1.** Microbial counts of presumptive thermophilic cocci and lactobacilli, mesophilic cocci and lactobacilli, enterococci, staphylococci, and coliforms in the frozen natural starter culture (NSC) and the lyophilized NSC (LNSC). Microbial counts were expressed as Log CFU/g $\pm$ standard deviation. **\***, significant difference ($p < 0.05$) in the Log CFU/g between NSC and LNSC, according to the Student *t* test.

### 3.2. Biodiversity Evaluation

### 3.2.1. Biodiversity at Species Level

A total of 157 isolates from the natural starter cultures NSC and LNSC (53 and 104, respectively) were molecularly identified at the species level. In particular, 135 isolates were identified by genus/species-specific PCR (36 from NSC and 99 from LNSC) and the remaining 22 isolates by 16S rRNA sequencing analysis (17 from NSC and 5 from LNSC).

The characterisation by genus/species-specific PCR revealed the presence of different microbial species in the two starter cultures (Figure 2). In particular, 25 *E. durans* isolates, only in the lyophilized culture (LNSC) in M17 (incubated at 22 and 45 °C) and MRS media, at dilution -6 were found. *E. faecium* was found in both cultures. In NSC, 10 isolates, all coming from the KAA medium at dilution -4, were found, whereas in LNSC, the *E. faecium* isolates characterised were 45: 23 from KAA at dilution -5, 17 from MRS at dilution -6, and 5 from M17 incubated at 22 and 45 °C, at dilution -6. Three *E. faecalis* were isolated, one from NSC (in KAA medium at dilution -4) and two from LNSC (in M17 incubated at 30 °C, at dilution -6). Furthermore, 35 colonies of *L. paracasei* were isolated from the FH medium, 12 from NSC, at dilution -3, and 23 from LNSC, at dilution -5. *S. gallolyticus* subsp. *macedonicus* was isolated more frequently from NSC than LNSC (13 versus 4 isolates, respectively), from the M17 medium at dilutions -5 and -6. Among streptococci, another four species were isolated and identified: 3 *S. oralis* (1 from NSC and 2 from LNSC) isolated from M17 at 45 °C, 1 *S. salivarius* isolated only from NSC in KAA at dilution -4, and 18 isolates belonging to the *S. bovis/equinus* complex (SBSEC), 15 from NSC and 3 from LNSC. For this latter microbial group, the *sod*A gene, encoding for the manganese-dependent superoxide dismutase, one of the most reliable biomarkers for SBSEC [29,30], was used for the identification at the species level. All 18 isolates were confirmed to belong to SBSEC; 13 of them were *S. equinus* (isolated from NSC in M17 30 °C and 45 °C, and MRS) and 5 were *S. lutetiensis* (2 from NSC and 3 from LNSC) (Figure 2).

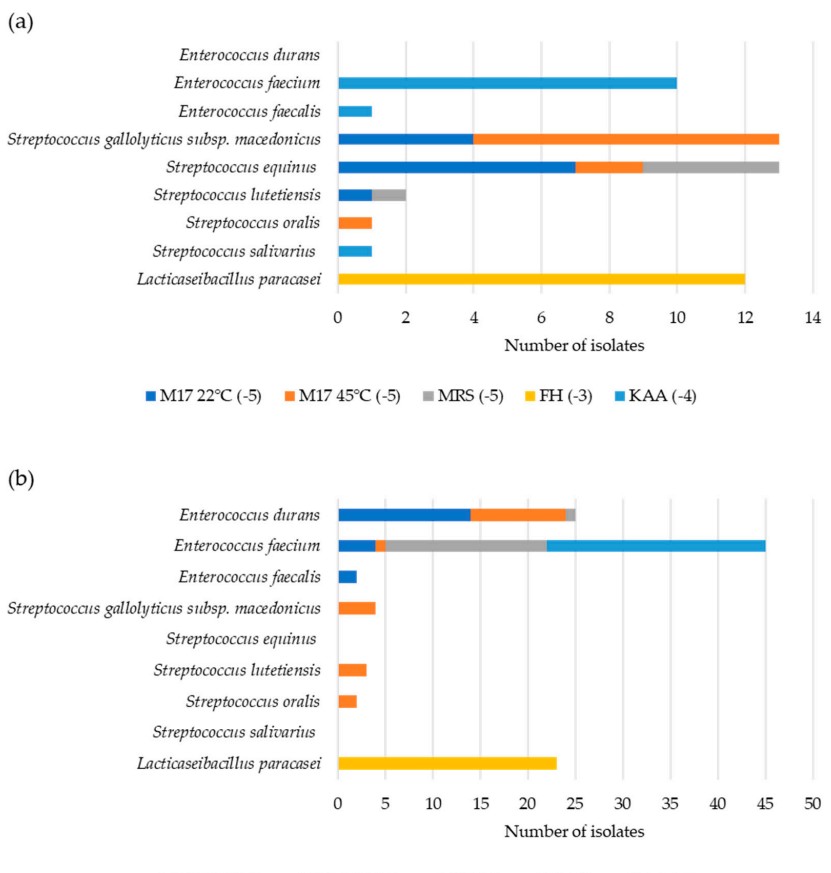

**Figure 2.** Bacterial species isolated from the natural cultures NSC (**a**) and LNSC (**b**). From each medium (M17 incubated at 22 °C, M17 incubated at 45 °C, MRS, FH, and KAA) and serial 10-fold dilution (in brackets), the numbers of isolates are indicated.

### 3.2.2. Biodiversity at Strain Level

The microbial isolates picked up from NSC and LNSC were also characterised at the strain level by (GTG)$_5$-rep and PFGE fingerprinting to calculate the number of microbial biotypes in the two forms of the natural starter culture investigated and to assess the biodiversity level. The calculation of the Simpson's diversity index (DI) revealed that *E. durans* (25 isolates), found only in LNSC (Table 2), was represented by eight rep-PCR profiles, with a DI of 0.85. The other *Enterococcus* species, *faecium* and *faecalis*, were instead found in both NSC and LNSC. In particular, six *E. faecium* biotypes were found in NSC (from 10 total isolates) and nine in LNSC (from 45 isolates), with one biotype in common between the two forms of the natural culture. Moreover, a DI decrease after the freeze-drying process, from 0.87 in NSC to 0.62 in LNSC, was observed. Two *E. faecalis* biotypes were calculated using the Pearson's correlation of (GTG)$_5$-rep fingerprints, one in NSC and one in LNSC, and the biotype found was in common between the cultures. However, DI calculation for *E. faecalis* was not possible since a single biotype was found both in NSC and LNSC. In addition, *L. paracasei* was also found both in NSC (12 isolates) and LNSC (23 isolates), with one and five biotypes detected, respectively. The DI for *L. paracasei* in LNSC was 0.78, whereas the diversity calculation was not possible in NSC. *S. gallolyticus* subsp. *macedonicus*, found both in NSC and LNSC, was represented by one and two biotypes, respectively, and the DI accountable, only for LNSC, was 0.50. Two *S. oralis* biotypes were found, one in NSC and one in LNSC, and none were in common, whereas only one *S. salivarius* from NSC was found. For the species belonging to the SBSEC complex, one molecular biotype of *L. lutetiensis* from NSC was found, and the same biotype was found in LNSC, whereas *S. equinus* was isolated only from NSC, and only one biotype (from 13 isolates) was found.

**Table 2.** Microbial biotypes for each species in the natural starter culture (NSC) and in the lyophilized NSC (LNSC).

| Species | NSC | | | LNSC | | | NSC + LNSC | | | |
|---|---|---|---|---|---|---|---|---|---|---|
| | Isolates No. | rep-PCR Profiles No. | PFGE Profiles No. | Isolates No. | rep-PCR Profiles No. | PFGE Profiles No. | Isolates No. | rep-PCR Profiles No. | rep-PCR Profiles in Common | PFGE Profiles in Common |
| *Enterococcus durans* | 0 | n.d. | n.d. | 25 | 8 | n.d. | 25 | 8 | n.d. | n.d. |
| *Enterococcus faecium* | 10 | 6 | n.d. | 45 | 9 | n.d. | 55 | 14 | 1 | n.d. |
| *Enterococcus faecalis* | 1 | 1 | n.d. | 2 | 1 | n.d. | 3 | 2 | 1 | n.d. |
| *Lacticaseibacillus paracasei* | 12 | 1 | n.d. | 23 | 5 | n.d. | 35 | 6 | n.d. | n.d. |
| *Streptococcus gallolyticus* subsp. *macedonicus* | 13 | 1 | n.d. | 4 | 2 | n.d. | 17 | 2 | 1 | n.d. |
| *Streptococcus oralis* | 1 | 1 | n.d. | 2 | 1 | n.d. | 3 | 2 | 0 | n.d. |
| *Streptococcus salivarius* | 1 | 1 | n.d. | 0 | n.d. | n.d. | 1 | 1 | 0 | n.d. |
| *Streptococcus lutetiensis* | 2 | 1 | 1 | 3 | 1 | 2 | 5 | 1 | 1 | 1 |
| *Streptococcus equinus* | 13 | 1 | 3 | 0 | n.d. | n.d. | 13 | 1 | 0 | 0 |
| Total isolates | 53 | | | 104 | | | 157 | | | |

Rep-PCR, (GTG)5 rep-PCR; PFGE, pulse field gel electrophoresis; n.d., not determined.



*3.3. Safety Assessment*

All of the 55 *E. faecium* isolates, 10 from NSC and 45 from LNSC, showing ampicillin MIC $\leq 2$ mg/L (Table 2) and absence of the virulence-associated genes $hyl_{Efm}$, *esp*, and *IS*16, were considered safe to be used as food additives since they met the standards of compliance with the safety requirements indicated by the EFSA [31].

Results about the safety of the other microbial species isolated are reported below.

3.3.1. Antibiotic Resistance and Virulence Genes Investigation

The cultures NSC and LNSC, *in toto*, and the microbial isolates belonging to the genus *Streptococcus*, were analysed for the detection of resistance genes for macrolides (*erm*A, *erm*B, *erm*C, and *mef*A), lincosamides (*lnu*C), tetracyclines (the putative conjugative Tn916-like transposon), and for the presence of potential virulence genes. *mef*A, encoding for an efflux pump, the activity of which, driven by proton motive force, involves only macrolides that determine the M phenotype [32] was detected in *S. oralis* and *S. salivarius*, both phenotypically resistant to azithromycin and erythromycin. The cultures NSC and LNSC, *in toto*, were positive for the Tn916-like transposon, whereas they, and the isolates, were negative for the antibiotic resistance genes listed above. LNSC cultures were negative for pathogenic-related genes $hyl_{Efm}$, *esp*, and *IS*16, while NSC was positive for *esp* due to the presence of *E. faecalis*.

3.3.2. Minimum Inhibitory Concentration Determination

The MIC of the antibiotics included in Table 3 was determined for all the isolates except those belonging to *E. faecium*. All of the 25 *E. durans* isolates (from LNSC) were susceptible to chloramphenicol, tetracycline, erythromycin, ampicillin, linezolid, tigecycline, vancomycin, teicoplanin, and ciprofloxacin (Table 3), using the cut off established by EUCAST or CLSI, as indicated in Table 3.

**Table 3.** Antibiotic susceptibility of *Enterococcus* isolates from the natural cultures.

| Bacteria Tested | Culture | Antibiotics Tested | | | | | | | | |
|---|---|---|---|---|---|---|---|---|---|---|
| | | Penicillins AMP | Macrolides ERY | Amphenicols CHL | Oxazolidinones LZD | Tetracyclines TET | Glycycyclines TGC | Glycopeptides VAN | TEI | Fluoroquinolones CIP |
| *E. faecium* | NSC | S [1] | n.t. | n.t. | n.t. | n.t. | n.t. | n.t. | n.t. | n.t. |
| | LNSC | S [1] | n.t. | n.t. | n.t. | n.t. | n.t. | n.t. | n.t. | n.t. |
| *E. faecalis* | NSC | S [1] | I [2] | S [2] | S [1] | R [2] | S [1] | S [1] | S [1] | S [1] |
| | LNSC | S [1] | I [2] | S [2] | S [1] | R [2] | S [1] | S [1] | S [1] | S [1] |
| *E. durans* | NSC | abs. | abs. | abs. | abs. | abs. | abs. | abs. | abs. | abs. |
| | LNSC | S [1] | S [2] | S [2] | S [1] | S [2] | S [1] | S [1] | S [1] | S [1] |

NSC, natural starter culture; LNSC, Lyophilized natural starter culture (LNSC). AMP, Ampicillin; PEN, Penicillin; AZI, Azithromycin; ERY, Erythromycin; FEP, Cefepime; FOT, Cefotaxime; AXO, Ceftriaxone; CHL, Chloramphenicol; CLI, Clindamycin; ETP, Ertapenem; MERO, Meropenem; LEVO, Levofloxacin; LZD, Linezolid; TET, Tetracycline; TGC, Tigecycline; VAN, Vancomycin; TEI, Teicoplanin; SYN, Quinupristin/Dalfopristin; CIP, Ciprofloxacin; S, sensitive; R, resistant; I, intermediate; n.t., not tested; abs., the species was absent in the culture. [1] Breakpoint by EUCAST 2023. [2] Breakpoint by CLSI 2020.

Among the enterococci investigated, the *E. faecalis* isolates, one from NSC and two from LNSC, were resistant to tetracycline and intermediate to erythromycin (CLSI cut off) [26] but susceptible to ampicillin, chloramphenicol, linezolid, tigecycline, vancomycin, teicoplanin, and ciprofloxacin (Table 3).

The 13 *S. gallolyticus* subsp. *macedonicus* isolated from NSC, and the 4 isolates from LNSC, were susceptible to penicillin, azithromycin, erythromycin, cefepime, cefotaxime, ceftriaxone, chloramphenicol, clindamycin, ertapenem, meropenem, levofloxacin, linezolid, tetracycline, and vancomycin. The same results were found for all the *S. lutetiensis* isolates (2 from NSC and 3 from LNSC) and the 13 *S. equinus* isolated only from NSC. *S. oralis* (one isolate from NSC and two from LNSC) was resistant to azithromycin and erythromycin. The only *S. salivarius* isolated from NSC was resistant to azithromycin and erythromycin, and it showed an intermediate profile for penicillin susceptibility (Table 4).

Table 4. Antibiotic susceptibility of *Streptococcus* isolates from the natural cultures.

| Bacteria Tested | Culture | Antibiotics Tested | | | | | | | | | | | | | | | |
| | | Penicillins | | Macrolides | | Cephalosporins | | | Amphenicols | Lincosamides | Carbapenems | | Fluoroquinolones | Oxazolidinones | Tetracyclines | Glycopeptides | Streptogramins |
| | | AMP | PEN | AZI | ERY | FEP | FOT | AXO | CHL | CLI | EPT | MERO | LEVO | LZD | TET | VAN | SYN |
| *S. gallolyticus macedonicus* | NSC | n.t. | S[2] * | S[2] | S[2] | S[2] | S[2] | S[2] | S[2] | S[2] | S[2] | S[2] | S[2] | S[2] | S[2] | S[2] | n.t. |
| | LNSC | n.t. | S[2] | S[2] | S[2] | S[2] | S[2] | S[2] | S[2] | S[2] | S[2] | S[2] | S[2] | S[2] | S[2] | S[2] | n.t. |
| *S. equinus* | NSC | n.t. | S[2] | S[2] | S[2] | S[2] | S[2] | S[2] | S[2] | S[2] | S[2] | S[2] | S[2] | S[2] | S[2] | S[2] | n.t. |
| | LNSC | abs. | abs. | abs. | abs. | abs. | abs. | abs. | abs. | abs. | abs. | abs. | abs. | abs. | abs. | abs. | abs. |
| *S. lutetiensis* | NSC | n.t. | S[2] | S[2] | S[2] | S[2] | S[2] | S[2] | S[2] | S[2] | S[2] | S[2] | S[2] | S[2] | S[2] | S[2] | n.t. |
| | LNSC | n.t. | S[2] | S[2] | S[2] | S[2] | S[2] | S[2] | S[2] | S[2] | S[2] | S[2] | S[2] | S[2] | S[2] | S[2] | n.t. |
| *S. oralis* | NSC | S[1] | S[2] | R[2] | R[2] | S[2] | S[2] | S[2] | S[2] | S[2] | S[2] | S[2] | S[2] | S[2] | S[2] | S[2] | I[2] |
| | LNSC | n.t. | S[2] | R[2] | R[2] | S[2] | S[2] | S[2] | S[2] | S[2] | S[2] | S[2] | S[2] | S[2] | S[2] | S[2] | n.t. |
| *S. salivarius* | NSC | S[1] | I[2] | R[2] | R[2] | S[2] | S[2] | S[2] | S[2] | S[2] | S[2] | S[2] | S[2] | S[2] | S[2] | S[2] | S[2] |
| | LNSC | abs. | abs. | abs. | abs. | abs. | abs. | abs. | abs. | abs. | abs. | abs. | abs. | abs. | abs. | abs. | abs. |

NSC, natural starter culture; LNSC, Lyophilized natural starter culture (LNSC). AMP, Ampicillin; PEN, Penicillin; AZI, Azithromycin; ERY, Erythromycin; FEP, Cefepime; FOT, Cefotaxime; AXO, Ceftriaxone; CHL, Chloramphenicol; CLI, Clindamycin; ETP, Ertapenem; MERO, Meropenem; LEVO, Levofloxacin; LZD, Linezolid; TET, Tetracycline; TGC, Tigecycline; VAN, Vancomycin; TEI, Teicoplanin; SYN, Quinupristin/Dalfopristin; CIP, Ciprofloxacin; S, sensitive; R, resistant; I, intermediate; n.t., not tested; abs., the species was absent in the culture. * one isolate was intermediate. [1] Breakpoint by EUCAST 2023. [2] Breakpoint by CLSI 2020.

## 4. Discussion

In this study, a natural starter culture, in frozen and lyophilized form, obtained from raw ewe's milk was investigated for its biodiversity and safety, at the strain level, with the perspective of use as a food additive in cheesemaking. The novelty of this work follows the new approach of Chessa et al. [11], where a new method for the obtainment of a natural starter culture directly from raw ewe milk without applying any thermal treatment or selection of pro-technological starter strains, to recover as much of the microbial raw milk biodiversity, was described. The evaluation of the microbial biodiversity was performed both on the frozen natural starter culture (NSC) and on its lyophilized form (LNSC). The latter is easier to handle in cheesemaking at both artisanal- and industrial-scale dairy plants. The bacterial isolates were picked up from the Petri dishes used for plate counts performed in the previous work and revealed the presence of eight species belonging to three genera, with different biotypes for each species. *L. paracasei* was the sole representative of the *Lacticaseibacillus* genus, with one biotype found in NSC and five in LNSC. The genus *Enterococcus* was represented by three species: *E. durans*, with eight strains found only in LNSC; *E. faecalis*, with one biotype detected both in the frozen NSC and in the lyophilized LNSC; and *E. faecium*, of which several biotypes were found (six in NSC and nine in LSNSC), with one of them shared between the cultures' forms. Enterococci are not currently included in the QPS list provided by EFSA every three years and updated every six months, and, therefore, their use as a food additive is not yet recommended [9] since they can be involved in nosocomial infections and antibiotic resistance spread [33]. That being said, their presence in cheese is desirable, due to their well-known contribution to aroma development [34]. Although infections caused by enterococci in humans, outside the nosocomial environment, are not common, the frequency of *E. faecium* isolated from hospitalised patients (i.e., belonging to the clade A) and responsible for infections has increased in the last decades [31]. To exclude the origin of the *E. faecium* strains from clade A, and, therefore, the possibility that they might have an advantage in the gastrointestinal tract given by ampicillin, amoxicillin, or vancomycin resistance, the evaluation of ampicillin susceptibility must be carried out. The EFSA suggested that *E. faecium* isolates having ampicillin MIC $\leq$ 2 mg/L and negative for the presence of the genes IS*16* (a marker associated with nosocomial strains), *esp* (pathogenicity marker), and $hyl_{Efm}$ (able to facilitate intestinal colonisation) can be considered safe and used as feed additives, assuming the microbial isolate tested is of environmental origin [31]. All the *E. faecium* biotypes investigated in this study complied with the requirements set by EFSA and can be safely used for cheesemaking. Although the assessment of safety for *E. faecium* is well described by the EFSA, this evaluation for other species is still not clear. Recently, the EFSA received notifications of re-assessment for the inclusion of several microorganisms in the QPS list. Among these, nine notifications for *E. faecium* were not evaluated, and this species is still outside the QPS list [8]. Moreover, other taxonomic units found in the starter culture investigated in this study have been evaluated for possible QPS status by the EFSA, concluding that *S. salivarius*, one isolate found only in NSC, is not recommended for the QPS list due to its ability to cause bacteraemia and systemic infection that results in a variety of morbidities [35], even though it is a commensal bacterium in the oral cavity and seems to contribute to human health, preventing biofilm formation [36]. Similarly, *S. oralis*, found both in NSC and LNSC, is a commensal species and opportunistic pathogen, showing low pathogenicity and virulence in immunocompromised patients [37]. Although its virulence mechanism is unclear, it is still not recommended for QPS status due to safety concerns [8]. Both *S. salivarius* and *S. oralis* were found in the starter cultures at dilution -4 for *S. salivarius* and at -5 and -6 for *S. oralis*; therefore, their presence in inoculated milk for cheesemaking is not that likely. Indeed, if the starter culture ($10^{10}$ Log CFU/mL) is inoculated at a final concentration of $10^5$ Log CFU/mL, up to 1 to 10 CFU/mL might be present in the inoculated milk. Presumably, their presence in ripened cheese and survival in the gastrointestinal tract after ingestion should not be a cause of concern. To gain a more detailed picture of these strains, for which the safety ascription is still of debate,

an antibiotic resistance evaluation was performed in this study. *S. salivarius* and *S. oralis* were sensitive to all the antibiotics tested except for macrolides, revealing resistance at low concentrations (2–4 mg/L) of erythromycin and azithromycin (M-phenotype, 14–15 membered ring macrolides) [32]. At the molecular level, they were positive to the *mef*A gene, coding for an efflux pump, that is correlated to the M-phenotype. High rates (76%) of erythromycin resistance were found in commensal isolates belonging to different sequence types by multilocus sequence typing [36]. Macrolide resistance in the *viridans* group of streptococci from healthy people's oropharynx is reported in the literature [38]. Moreover, the other genes tested for macrolide (*erm*B) and lincosamide (*lnu*C) resistance were absent.

The other *Streptococcus* found in the culture were *S. gallolyticus* subsp. *macedonicus*, *S. equinus*, and *S. lutetiensis*, which are included in the *S. bovis-S. equinux* complex (SBSEC), the non-enterococcal group D streptococci, and, from a safety point of view, could deserve attention [39]. *S. gallolyticus* subsp. *macedonicus*, commonly found in several European cheeses, was isolated for the first time from naturally fermented Greek Kasseri cheese, and it can also be found in Italian cheeses [40]. It is moderately acidifying and proteolytic, potentially contributing to cheese ripening, and can be considered a multifunctional candidate as a nonstarter lactic acid bacterium and adjunct culture for dairy manufacturing since it is non-pathogenic [41]. *S. equinus* and *S. lutetiensis*, inhabitants of the rumen and gastrointestinal tracts of animals and humans, are associated with bovine mastitis [42]. Although considered potential human pathogens [43], they have been isolated from camel, buffalo, and bovine milk, and also from traditional fermented milks from Gambia and Ethiopia, obtained after natural fermentation of raw cow and camel milks [44,45], and from the traditional cheeses Darfiyeh, a Lebanese product from raw goat milk [46], and Mozzarella di Bufala Campana, an Italian protected designation-of-origin (PDO) cheese [47]. Similar to the *Enterococcus* strains isolated from the natural culture, all the *S. gallolyticus* subsp. *macedonicus* and *S. lutetiensis* were negative for antibiotic resistance at the molecular and phenotypic level and, since these species are commonly found in raw milks [33,48], it is believed that they can be used for cheesemaking, even if not included in the QPS list.

Potential pathogens can be introduced by food consumption, and their survival in the human and animal GIT is affected by diet. Therefore, oral, pyogenic, and other streptococci, like those causing mastitis, can be considered food-related microorganisms. The presence of streptococci such as *S. equinus* are considered to be indicators of faecal pollution of food because they have an advantage over coliforms as they are more resistant to most environmental stresses [41]. Therefore, the results of this study show that some *Streptococcus* and *Enterococcus* species were found in the natural starter culture investigated. Indeed, as described by Chessa et al. [11], the culture was obtained from raw milk without applying any thermal treatment but only a slightly acidic condition to remove potential pathogens, and spoilage and hygiene indicators, such as coliforms and staphylococci. Indeed, resilient microorganisms like streptococci or acidic-tolerant like enterococci, already present in the raw milk, survived and were detected in both the natural starter culture NSC and also, after lyophilisation, in LNSC. The potential pathogens found, *E. faecium* and the *Streptococcus* species, were present in the raw ewe milk used in this study and consequently found in the natural starter culture. Furthermore, some SBSEC members such as *S. gallolyticus* subsp. *macedonicus* and *S. lutetiensis* are part of the daily diet, also producing bacteriocins useful for food preservation [49], and can be considered safe, although some strains may be potentially pathogenic [39].

The natural culture investigated, both in frozen and lyophilized form, revealed good biodiversity, both at the species and strain level. Unlike commercial starters, built by few selected species/strains, usually one to three, this was obtained directly from raw ewe milk without any thermal treatment nor isolation to select pro-technological bacteria. Nevertheless, some species found were potentially pathogens and marked for attention by the EFSA as they are not yet included in the QPS list, thus requiring attention before being used as food additives. Nonetheless, all the tests, both at molecular and phenotypic levels, gave reassurances regarding their safety and suggest their suitability for cheesemaking.

## 5. Conclusions

The obtainment of a natural culture directly from raw ewe milk bypassing the thermal treatment and the selection of pro-technological bacteria may be advantageous in terms of microbial diversity. Conversely to the commercial starters composed of a few species and/or strains cultured separately then artificially mixed together, the natural culture is characterised by a variety of species and strains in a delicate equilibrium that can contribute to the uniqueness and typicity of artisanal products. The assessment of taxonomic identity of microorganisms was the first step in the evaluation of the microbial composition, at the strain level, but this work also aimed to focus attention in terms of food safety since non-QPS microorganisms can be included in the natural starter and can be found in cheeses, especially in traditional ones obtained from raw milk, where thermal treatment is not commonly applied or even not allowed by strict production regulations, like those for the PDOs. The current QPS list does not include some microbial species commonly present in raw milk, inevitably found in traditional products and even in the natural starter culture investigated. Nonetheless, it contained the most important characteristics in addition to technological abilities, i.e., biodiversity and safety. Following the actual EFSA guidelines, artisanal factories may not be able to provide the production of starter cultures from raw milk by themselves, running the risk of including some non-QPS species in their culture, and only commercial selected starters would be allowed for cheese production. Considering the findings of this study, a revision of the criteria for accession to the QPS list should be performed.

**Supplementary Materials:** The following supporting information can be downloaded at: https://www.mdpi.com/article/10.3390/fermentation10010029/s1, Table S1: Origin of the NSC and LNSC bacterial isolates picked out from the 10-fold dilution of the respective media; Table S2: Molecular primers used for antibiotic resistance genes and virulence genes detection; Table S3: Breakpoints used for the safety evaluation of non-QPS microorganisms.

**Author Contributions:** Conceptualization, L.C. and R.C.; methodology, L.C., A.P. and R.C.; formal analysis, L.C. and I.D.; investigation, E.D., I.D., M.C.F. and D.G.D.; data curation, L.C., E.D. and I.D.; writing—original draft preparation, L.C.; writing—review and editing, L.C. and R.C.; project administration, R.C.; funding acquisition, R.C. All authors have read and agreed to the published version of the manuscript.

**Funding:** This research was funded by the Regione Autonoma della Sardegna—Progetto ValIdeS—Valorizzazione e tutela dei sistemi di produzione agroalimentare Identitari del centro Sardegna—L.R.7/2007, "Promozione della ricerca scientifica e dell'innovazione tecnologica in Sardegna" annualità 2020.

**Institutional Review Board Statement:** Not applicable.

**Informed Consent Statement:** Not applicable.

**Data Availability Statement:** Data are contained within the article and Supplementary Materials.

**Conflicts of Interest:** The authors declare no conflicts of interest.

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
