# Peer review of "Biodiversity and Safety: Cohabitation Experimentation in Undefined Starter Cultures for Traditional Dairy Products"

_fermentation, doi:10.3390/fermentation10010029_

Round 1

Reviewer 1 Report

Comments and Suggestions for Authors

This manuscript titled “Biodiversity and safety: cohabitation experimentation in undefined starter cultures for traditional dairy products” (No. fermentation-2768594) discussed the diversity and safety of frozen and lyophilized natural starter cultures which are involved in dairy products at molecular and phenotypic levels. The work is meaningful for the processing of cheesemaking in giving reassurances regarding some species that were potentially pathogens safety, such as S. equinus, and thus enlarging the QPS list. In addition, the experiments designed for this study were performed well with a lot of data. However, some drawbacks including errors still exist in the present version of this manuscript. Thus, authors have to make a major revision to their manuscript before consideration for acceptance.

Specific points are issued as follows.

1/Line 48, “technological performance” means what? Hard to understand.

2/Line 53-54, not clear, please give some examples to illustrate which strains are good or which have a potential risk for people.

3/What logic connection between line 64-65 and line 66-67?

4/The part Introduction needs further improvement with the justification of the novelty of undefined starter cultures for traditional dairy products.

5/For the section “2.4.1 Antibiotic resistance and virulence genes detection”, how to prepare the samples for “the pathogenic-related genes hylEfm, esp, and IS16, tetracycline resistance genes, and antibiotic resistance genes”?

6/Line 172-175, how much volume of 7 E. faecium strains used? And “positive and negative control wells were also included” indicates what, not clear.

7/Line 209, “2.2 and 1.2 Log CFU/g of culture, respectively” not clear.

8/Line 228, “M17 (incubated at 22 and 45°C)”, but Figure 2 only includes “M17 incubated at 30°C, and M17 incubated at 45°C”, thus “22°C” is right? If right, please display relevant results.

9/The part Conclusions should add some results of this manuscript instead of only specify the significant of its study.

10/For the part References, journal name should be unified for their formats. And the Latin name for species should be in italic.

Author Response

Reply to the Reviewer 1

1/Line 48, “technological performance” means what? Hard to understand.

Generally, when speaking about starter cultures, technological performance is referred to their acidification ability. The sentence in the manuscript was improved.

2/Line 53-54, not clear, please give some examples to illustrate which strains are good or which have a potential risk for people.

The paragraph was reworded following the Reviewer’s suggestions.

3/What logic connection between line 64-65 and line 66-67?

Line 66-67 were moved and are now in lines 57-59.

4/The part Introduction needs further improvement with the justification of the novelty of undefined starter cultures for traditional dairy products

Actually, the use of a natural starter culture for traditional dairy products is not a novelty. The text was improved as suggested to make it clearer.

5/For the section “2.4.1 Antibiotic resistance and virulence genes detection”, how to prepare the samples for “the pathogenic-related genes hylEfm, esp, and IS16, tetracycline resistance genes, and antibiotic resistance genes”?

The paragraph was improved according to the Reviewer’s comments.

6/Line 172-175, how much volume of 7 E. faecium strains used? And “positive and negative control wells were also included” indicates what, not clear.

The paragraph was improved according to the Reviewer’s comments.

7/Line 209, “2.2 and 1.2 Log CFU/g of culture, respectively” not clear.

The sentence was rephrased.

8/Line 228, “M17 (incubated at 22 and 45°C)”, but Figure 2 only includes “M17 incubated at 30°C, and M17 incubated at 45°C”, thus “22°C” is right? If right, please display relevant results.

22°C is the right temperature. The authors changed the Figure 2 and modified the temperature in the legend and in the caption.

9/The part Conclusions should add some results of this manuscript instead of only specify the significant of its study.

The authors considered writing what had already been reported and discussed in the “Results” and “Discussion” sections could be repetitive and useless, preferring to focus the reader's attention on the interpretation of results.

The authors improved the section adding the sentence: “Nonetheless, it contained the most important characteristics besides technological abilities, i.e. biodiversity and safety.”

10/For the part References, journal name should be unified for their formats. And the Latin name for species should be in italic.

The authors used the bibliography software package EndNote to handle the references in the manuscript, following the guidelines of the Journal. The Journals formats have been unified and the species names modified in italics manually, according to the Reviewer’s suggestions.

Reviewer 2 Report

Comments and Suggestions for Authors

- Please use full forms instead of abbreviations when presented for the first time in the manuscript. For example, EFSA, PFGE, and MIC in the abstract section. It is generally encouraged to limit the use of abbreviations in the abstract.

Comments on the Quality of English Language

-Minor English editing would improve the overall readability of the manuscript. 

Author Response

Reply to the Reviewer 2

Comments and Suggestions for Authors

- Please use full forms instead of abbreviations when presented for the first time in the manuscript. For example, EFSA, PFGE, and MIC in the abstract section. It is generally encouraged to limit the use of abbreviations in the abstract.

The abstract was revised as suggested.

Comments on the Quality of English Language

-Minor English editing would improve the overall readability of the manuscript.

The manuscript was thoroughly revised by a native English speaker.

Reviewer 3 Report

Comments and Suggestions for Authors

This manuscript authored by Chessa et al., deals with the evaluation of biodiversity, at species and strain level, of an undefined natural starter culture, obtained from raw ewe’s milk avoiding any thermal treatment or microbial selection, and to assess the safety of non-QPS strains in the perspective of use this culture for cheesemaking.

The manuscript is comprehensive, structurally and scientifically sound with proper flow. However, there are some issues:

1    Some minor corrections concerning keywords are needed. For example, I do not think that European Food and Safety Authority needs to be included.

S   Some mistakes like lyophilized instead of lyophilised should be corrected. In fact, English usage is substandard and impedes proper reading and understanding. Revision by a proficient scientific English translator is recommended.

Comments on the Quality of English Language

English usage is substandard and impedes proper reading and understanding. Revision by a proficient scientific English translator is recommended.

Author Response

Reply to the Reviewer 3

Comments and Suggestions for Authors

This manuscript authored by Chessa et al., deals with the evaluation of biodiversity, at species and strain level, of an undefined natural starter culture, obtained from raw ewe’s milk avoiding any thermal treatment or microbial selection, and to assess the safety of non-QPS strains in the perspective of use this culture for cheesemaking.

The manuscript is comprehensive, structurally and scientifically sound with proper flow. However, there are some issues:

1    Some minor corrections concerning keywords are needed. For example, I do not think that European Food and Safety Authority needs to be included.

“European Food and Safety Authority” was removed and the keywords were improved as suggested.

2   Some mistakes like lyophilized instead of lyophilised should be corrected. In fact, English usage is substandard and impedes proper reading and understanding. Revision by a proficient scientific English translator is recommended.

The authors changed lyophilised into lyophilized, and the manuscript was thoroughly revised by a native English speaker.

Comments on the Quality of English Language

English usage is substandard and impedes proper reading and understanding. Revision by a proficient scientific English translator is recommended.

Round 2

Reviewer 1 Report

Comments and Suggestions for Authors

No more comments